# Moderate and intensive mechanical loading differentially modulate the phenotype of tendon stem/progenitor cells *in vivo*

**Jianying Zhang**[1], **Daibang Nie**[1,2], **Kelly Williamson**[1], **Arthur McDowell**[1,3], **MaCalus V. Hogan**[1], **James H-C. Wang**[1,4,5] *

**1** MechanoBiology Laboratory, Department of Orthopaedic Surgery, University of Pittsburgh, Pittsburgh, PA, United States of America, **2** Department of Immunology, College of Basic Medicine, Chongqing Medical University, Chongqing, China, **3** Howard University College of Medicine, Washington D.C., United States of America, **4** Department of Bioengineering, University of Pittsburgh, Pittsburgh, PA, United States of America, **5** Department of Physical Medicine and Rehabilitation, University of Pittsburgh, Pittsburgh, PA, United States of America

* wanghc@pitt.edu

**Data Availability Statement:** All relevant data are within the manuscript and its Supporting Information files.

## Abstract

To examine the differential mechanobiological responses of specific resident tendon cells, we developed an *in vivo* model of whole-body irradiation followed by injection of either tendon stem/progenitor cells (TSCs) expressing green fluorescent protein (GFP-TSCs) or mature tenocytes expressing GFP (GFP-TNCs) into the patellar tendons of wild type C57 mice. Injected mice were subjected to short term (3 weeks) treadmill running, specifically moderate treadmill running (MTR) and intensive treadmill running (ITR). In MTR mice, both GFP-TSC and GFP-TNC injected tendons maintained normal cell morphology with elevated expression of tendon related markers collagen I and tenomodulin. In ITR mice injected with GFP-TNCs, cells also maintained an elongated shape similar to the shape found in normal/untreated control mice, as well as elevated expression of tendon related markers. However, ITR mice injected with GFP-TSCs showed abnormal changes, such as cell morphology transitioning to a round shape, elevated chondrogenic differentiation, and increased gene expression of non-tenocyte related genes LPL, Runx-2, and SOX-9. Increased gene expression data was supported by immunostaining showing elevated expression of SOX-9, Runx-2, and PPARγ. This study provides evidence that while MTR maintains tendon homeostasis by promoting the differentiation of TSCs into TNCs, ITR causes the onset of tendinopathy development by inducing non-tenocyte differentiation of TSCs, which may eventually lead to the formation of non-tendinous tissues in tendon tissue after long term mechanical overloading conditions on the tendon.

## Introduction

Tendons function to transmit mechanical loads from the muscle to bone, stabilizing the joint and enabling its motion. While normal mechanical loading maintains tendon homeostasis, excessive mechanical loading can cause chronic tendon injury or tendinopathy [1], which is a

**Funding:** All the external funding sources of support received during this study was from the National Institute of Arthritis and Musculoskeletal and Skin Diseases (NIAMS), NIH under awards AR065949 and AR070340 (JHW). The funders had no role in study design, data collection and analysis, decision to publish, or preparation of the manuscript. There was no additional external funding received for this study.

**Competing interests:** The authors have declared that no competing interests exist.

prevalent tendon disorder that affects millions of Americans and costs billions of healthcare dollars every year [2, 3]. The typical cellular and molecular alterations associated with tendinopathic tendons include an increase in proteoglycan deposition, extracellular matrix (ECM) degradation, and the rounding of cell nuclei, and the acquisition of chondrocyte phenotypes [4–9]. These abnormal cellular and molecular alterations in response to mechanical overloading may be brought about by either migrating cells, including non-tenocytes and/or bone marrow mesenchymal stem cells (BMSCs), or resident tendon cells, including tendon stem/progenitor cells (TSCs), which possess multilineage differentiation potential, and mature tenocytes (TNCs). However, the ability to successfully distinguish between migrating and resident cell populations *in vivo* is difficult with current techniques [10].

Using an *in vitro* model, we previously showed that one of the causes of tendinopathy development may be via the aberrant differentiation of TSCs into non-tenocytes when subjected to excessive mechanical loading [11]. Stem cells, especially within musculoskeletal tissues, are constantly modulated by mechanical loading, which may also induce aberrant differentiation of TSCs leading to the tendinopathic characteristics listed above [12, 13]. Chondrocyte markers are expressed in the clinical samples of calcific insertional Achilles tendinopathy [6] and rotator cuff tendinopathy [14, 15]. Our study also showed that under identical excessive conditions *in vitro*, isolated tendon cells or mature tenocytes did not differentiate into non-tenocytes [13]. Thus, aberrant TSC differentiation may *initiate* the formation of non-tendinous tissues in tendons that ultimately deter the normal tendon function and later cause tendon degeneration [11, 13, 16, 17].

Previously, we have shown the beneficial effects of low-level mechanical stretching in cultured mouse TSCs, and in TSCs isolated from mice after moderate treadmill running (MTR). Specifically, we have shown that a low level of mechanical stretching *in vitro* promoted the differentiation of TSCs into TNCs, and that MTR has overall beneficial effects by expanding the pool of total TSCs and in increasing TSC-related production of collagen, the major component of tendons [11, 18]. However, mechanical overloading in the form of intensive treadmill running (ITR) in mice resulted in long-term detrimental effects that can result in chronic inflammation and degenerative tendinopathy [11, 13, 19]. We also showed that large magnitude stretching of TSCs *in vitro* and ITR in mice *in vivo* induced aberrant differentiation of TSCs into non-tenocytes while TNCs remained unchanged [11, 13]. However, certain issues remain in using *in vitro* models because they exclude the cellular matrix and soluble factors, and thus impact the mechanism of tendinopathy development. On the other hand, *in vivo* models cannot discern whether migrating or resident cells contribute to the development of tendinopathy. Therefore, a novel *in vivo* model is necessary to study the roles of resident TSCs and TNCs in the development of tendinopathy, as well as in the maintenance of tendon homeostasis. Thus, we generated an *in vivo* model with an irradiation and injection approach in which we removed native tendon cells in wild type mice by irradiation, and then injected either TSCs expressing green fluorescent protein (GFP-TSCs) or TNCs expressing GFP (GFP-TNC) into the patellar tendons of irradiated mice that can be easily tracked through the expression of GFP.

Based on our previous studies, we hypothesized that ITR would induce TSCs, but not TNCs, to differentiate into non-tenocytes. To test this hypothesis, we subjected the irradiated mice injected with either GFP-TSCs or GFP-TNCs to both MTR and ITR. We show that only TSCs responded differently to MTR and ITR. In response to ITR, TSCs expressed non-tenocyte related genes and differentiated into chondrocytes. With MTR, TSCs expressed tenocyte-related genes thus promoting tendon maintenance and homeostasis by MTR. In contrast, TNCs responded to both MTR and ITR with elevated expression of tendon related markers only. We present a detailed report below.

## Methods

### Animals used for the experiments

We harvested GFP-tendon stem cells (GFP-TSCs) and GFP-tenocytes (GFP-TNCs) from the patellar tendons of 10 weeks old young adult GFP C57BL/6 transgenic mice (GFP-mice) [C57BL/6-TgN (ACTbEGFP)1Osb mice, The Jackson Laboratory, Bar Harbor, ME]. Isolated cells, either GFP-TSCs or GFP-TNCs, were separately injected into the patellar tendons of irradiated adult C57BL/6J mice (10 weeks old, female, The Jackson Laboratory, Bar Harbor, ME) and were utilized as described below. A total of 59 mice were utilized for this study as shown in Scheme (S1 Fig). The use of all animals was approved by the IACUC (#17019968) of the University of Pittsburgh.

### Isolation of GFP-TSCs and GFP-TNCs

TSCs and TNCs were isolated from five GFP-mice according to our previously published protocol [20] for injection and analysis prior to injection. The TSCs were isolated from individual cell colonies which were detached by local application of trypsin under microscopic visualization. The detached cells (TSCs) from each colony were collected using a micropipette and transferred to individual cell culture containers for further culture. After removal of the TSC colonies, elongated cells remained in the culture plates. These cells, presumably TNCs, were cultured further with the addition of 10% FBS in DMEM with 100 U/ml penicillin and 100 μg/ml streptomycin. For transplantation experiments, low passage cultures of TSCs at passage 2 were utilized to ensure their 'stemness' and ability to differentiate into tenocytes. Passage 2 TSCs were collected from individual cell cultures and mixed for transplantation use. For TNCs, passage 5 cells were used for transplantation to ensure these cells had fully differentiated into tenocytes.

### Cell morphology assessment and nucleostemin staining

The morphology of GFP-TSCs and GFP-TNCs was examined under a fluorescent microscope (Nikon eclipse, TE2000-U) before transplantation, and the stemness of isolated GFP-TSCs and GFP-TNCs was determined by immunostaining with the stem cell marker nucleostemin (NS). To perform immunostaining [20, 21], the GFP-TSCs or GFP-TNCs were separately seeded into 12-well plates at a density of $3.5 \times 10^4$/well and cultured with a growth medium for 3 days. Then the medium was removed, and the cells were washed once with PBS. The washed cells were fixed with PBS-buffered 4% paraformaldehyde for 20 minutes and treated with 0.1% Triton-X-100 for another 15 minutes. After washing the cells with PBS three times, the cells were incubated at 4°C with goat anti-nucleostemin overnight (1:350, Neuromics, Edina, MN). Then the cells were washed with PBS 3 times and incubated with cyanine 3 (Cy3)-conjugated donkey anti-goat IgG secondary antibody (1:500, Millipore, Temecula, CA) at room temperature for 1 hour. Finally, the cells were counterstained with Hoechst 33342 (1 μg/ml, Sigma, St Louis). The NS positive cells were determined under a fluorescent microscope.

### Irradiation, injection, and treadmill running

To determine the fate of injected GFP tendon cells (TSCs and TNCs) *in vivo*, we first eliminated native tendon cells by using whole body irradiation on a total of 48 adult C57BL/6J mice with 6 Gy dosage using the Gamma cell 40 irradiator (Best Theratronics, Ontario, Canada). The 6 Gy dosage was selected based on published literature [22, 23]. Briefly, mice were whole-body exposed to 6 Gy of gamma radiation without anesthesia in a specially designed well-ventilated mouse pie cage (Braintree Scientific Inc., Braintree, MA). Animals were irradiated in

batches of ten. Mice were monitored daily for the development of symptoms of radiation sickness after irradiation for the first seven days, as well as regular monitoring during subsequent treadmill running protocols. The effect of irradiation on tendon cells was confirmed using the live/dead cell viability assay kit (ThermoFisher Scientific, Waltham, MA) according to the manufacturer's protocol.

A total of 48 mice underwent irradiation and were utilized for separate experiments as follows. A total of three irradiated only wild type mice were utilized for irradiation testing by live/dead assay kit, and three irradiated only (6 Gy) mice were utilized for histological analysis by H&E staining as controls. For GFP-TSC and GFP-TNC injection, a total of 36 irradiated wild type mice were equally separated into three groups—cage, ITR, and MTR, with 12 mice per group. For each of these three groups (cage, ITR, and MTR), the patellar tendons were injected with one of the following treatments using a 30G syringe 48 hrs post-irradiation: **group 1** with 6 mice that received GFP-TSCs (P1, $1 \times 10^4$/μl, 50 μl/each tendon), or **group 2** with 6 mice that received GFP-TNCs (P5, $1 \times 10^4$/μl, 50 μl/each tendon). For these 36 irradiated and injected mice, a single injection was performed parallel to the tendon tissue from the paratenon to the center of the patellar tendon. For each mouse, both patellar tendons were injected. One week after injection, the 36 irradiated and injected mice were subjected to treadmill running protocols as described in previous studies [13, 18, 19]. In addition, 3 irradiated wild type mice with GFP-TSCs and 3 irradiated wild type mice with GFP-TNCs were analyzed to verify the migration and spreading of injected GFP cells using the live/dead cell viability assay kit (ThermoFisher Scientific, Waltham, MA) and immunohistochemistry. Finally, an additional 6 wild type untreated 'normal' control mice were utilized, specifically 3 mice for cell viability assay and 3 mice for histological analysis. Thus, overall, we utilized a total of 59 mice, with 48 mice receiving whole body irradiation and 6 untreated/wild type mice, and 5 GFP wild type mice used as controls.

Running protocols for GFP-TSC and GFP-TNC mice in either MTR or ITR groups were identical. All treadmill running mice in both MTR and ITR groups ran 15 min/day at 13 m/min for 5 days in the first week. From the second week, the MTR mice ran at the same speed for 50 min/day, 5 days/week for 3 weeks. The ITR mice ran at the same speed for 3 hrs/day, 4 hrs/day, and 5 hrs/day for 5 days/week in the second, third, and fourth weeks, respectively. The mice in the cage control group were allowed to move freely in the cage during treadmill running experiments. At the end of treadmill running, mice were sacrificed, the patellar tendon tissues were collected from each group. Three mice (6 patellar tendons) were used for histological analyses and 3 mice (6 patellar tendons) were used for gene expression analyses [11, 13, 19].

### Histological analysis of mouse tendon sections

To assess the response of injected GFP-TSCs and GFP-TNCs to mechanical loading *in vivo*, the patellar tendon tissues were collected from each group after irradiation, injection, and treadmill running. Tissues were then prepared for cryo-sectioning by freezing at -80˚C in Neg 50 filled molds (Richard-Allan Scientific: Kalamazoo, MI). Each frozen tissue block was cut into 10 μm thick sections and left to dry overnight at room temperature on glass slides. Tissue sections of GFP+ tendon cells in each group were observed through a fluorescent microscope and confirmed by immunostaining with rabbit anti-GFP primary antibody (Cat # 2956S, Cell Signaling Technology Inc, Danvers, MA), followed by Cy3-conjugated goat anti-rabbit IgG as the secondary antibody (Cat # AP132C, Millipore Sigma, Burlington, MA). The irradiation effect on tendon cells was detected with a live/dead cell viability assay kit (Cat # L-7013, Life Technologies, Carlsbad, CA) performed according to the manufacturer's protocols. The extent of cell viability in the injected patellar tendon area of each group was determined using semi-quantification methods described below.

For histochemical staining, fixed tissue sections were stained with hematoxylin and eosin (H&E), and Alcian blue and nuclear fast red according to our previous protocol [19]. For immunostaining analysis of non-tenocyte differentiation, the fixed tissue sections were further treated with 0.1% of Triton X-100 for 30 min at room temperature, then incubated with rabbit anti-SOX-9 (1:500, Millipore, Cat #AB5535, Billerica, MA), rabbit anti-Runx2 (1:200, Santa Cruz Biotechnology, Inc., Cat #sc-10758, Dallas, TX), and rabbit anti-PPARγ (1:300, Santa Cruz Biotechnology, Inc., Cat #sc-7196, Dallas, TX) antibodies at 4˚C overnight. Then, the tissue sections were washed three times with PBS, and incubated with Cy-3-conjugated goat anti-rabbit IgG as the secondary antibody (1:500, Cat # AP132C, Millipore Sigma, Burlington, MA) at room temperature for 1 hour. The positively stained cells were determined by a fluorescent microscope (Nikon Eclipse, TE2000-U, Japan).

## Semi-quantification of positively stained cells

The positively stained cells in each image were analyzed by semi-quantification. Five random images from each well and three random images from each tissue section at the same magnification were taken under a microscope (Nikon eclipse, TE2000-U, Melville, NY). At least three tissue sections from each group were examined from each marker. The positively stained cells in each picture were manually identified and analyzed using Spot imaging software (Diagnostic Instruments, Inc., Sterling Heights, MI). The percentage of positive staining was calculated by dividing the number of positively stained cells by the total number of cells under the microscopic field.

## Gene expression analysis of mouse tendon tissue samples

To investigate the mechanobiological response of GFP tendon cells injected into the patellar tendons of the mice subjected to treadmill running, we determined the expression of tenocyte and non-tenocyte related genes in the patellar tendon tissues of each group of mice using qRT-PCR according to our published protocol [13]. A total of six mice, or twelve patellar tendons, were sacrificed for each group cage, ITR, and MTR, which included three GFP-TSC mice and three GFP-TNC mice per group.

Previously published mouse-specific primers were used for the following tenocyte-related genes, collagen type I and tenomodulin [24] and non-tenocyte-related genes, lipoprotein lipase (LPL) (adipocyte-related genes), SOX-9 (chondrocyte-related gene), and Runx2 (osteocyte-related gene) [25–27]. Glyceraldehyde-3-phosphate dehydrogenase (GAPDH) was used as an internal control [24]. All primers were obtained from Invitrogen (Grand Island, NY) and their sequences are shown in **Table 1**. PCR reaction conditions and calculation of gene expression levels were as previously described [11]. The gene expression levels in each tendon tissue sample were calculated with the formula $2^{-\Delta\Delta CT}$, where $\Delta\Delta CT = (CT_{target} - CT_{GAPDH})_{treated} - (CT_{target} - CT_{GAPDH})_{control}$, and CT represented the cycle threshold of each RNA sample. To calibrate gene expression data, cage control mice were injected with the same cells as the control, and gene expression of MTR or ITR mice was compared to the cage control using the $2^{-\Delta\Delta CT}$ method. Each sample was normalized to their respective GAPDH expression. At least three separate experiments were performed to determine the standard deviation (SD) of the ΔCT.

## Statistical analysis

All data represents mean ± SD from at least three replicates. A student's *t*-test was used for the comparison of two groups. For more than two groups, one-way ANOVA was used, followed by Fisher's least significant difference (LSD) test for multiple comparisons with statistical significance set at $p < 0.05$.

**Table 1. Mouse primer sequences for qRT-PCR.**

| Gene | Primer sequence (forward) | Primer sequence (Reverse) | Reference |
|---|---|---|---|
| Collagen I | 5'-CCA GCG AAG AAC TCA TAC AGC-3' | 5'-GGA CAC CCC TTC TAC GTT GT-3' | Mendias C. et al. |
| Tenomodulin | 5'-TGT ACT GGA TCA ATC CCA CTC T-3' | 5'-GCT CAT TCT GGT CAA TCC CCT-3' | Mendias C. et al. |
| LPL | 5'-AAG CTG GTG GGA AAT GAT GTG G-3' | 5'-CCG TTC TGC ATA CTC AAA GTT AGG-3' | Ruge T. et al. |
| Runx-2 | 5'-CGA CAG TCC CAA CTT CCT GT-3' | 5'-CGG TAA CCA CAG TCC CAT CT-3' | Teplyuk N. et al. |
| SOX-9 | 5'-GCT TGA CGT GTG GCT TGT TC-3' | 5'-GAG CCG GAT CTG AAG ATG GA-3' | Xiong Y. et al. |
| GAPDH | 5'-TGG AAA GCT GTG GCG TGA T-3' | 5'-TGC TTC ACC ACC TTC TTG AT-3' | Mendias C. et al. |

## Results

### The differential characteristics of isolated GFP tendon cells in vitro

GFP-TSCs and GFP-TNCs were isolated from the patellar tendons of five GFP-mice and analyzed prior to injection. The morphological analysis showed that GFP-TSCs have a cobblestone shape with strong green fluorescence (Fig 1A), with semi-quantitation showing 89% of these

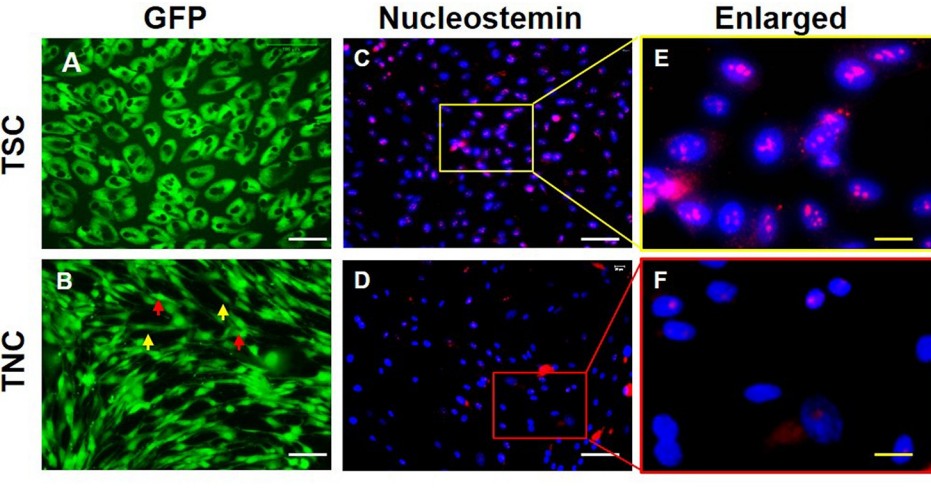

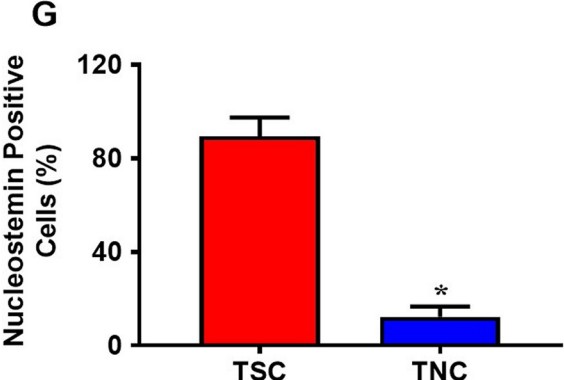

**Fig 1. The properties of TSCs differ from that of TNCs isolated from GFP mouse patellar tendons.** TSCs exhibit cobble stone shape (**A**) and TNCs are a mix of elongated and spindle-like cells (yellow and red arrows, **B**). Immunostaining shows that nucleostemin (NS) is highly expressed in TSCs (**C, E**), but NS expression is much lower in TNCs (**D, F**). The images of **E** and **F** are enlarged images of the boxes in **C** and **D**, respectively. Semi-quantification of positively stained cells shows that 89% of TSCs express NS and about 12% of TNCs express NS (**G**). Graph bars represent the mean ± SD. *p < 0.05 compared to TSCs. White bars: 100 μm; yellow bars: 25 μm.

cobblestone-shaped cells positively stained with the stem cell marker nucleostemin (**Fig 1C, 1E and 1G**). After removing GFP-TSC colonies from the primary culture plate, the remaining cells were further cultured to passage 5, yet cells were unable to form colonies. At passage 5, these cells showed typical tenocyte morphology of an elongated shape (yellow arrows, **Fig 1B**), and some of them were spindle-shaped (red arrows, **Fig 1B**). Both cells with elongated and spindle morphology still produced a strong green fluorescence, but only 12% of GFP-TNCs expressed nucleostemin (**Fig 1D, 1F and 1G**). Based on the cell shape and loss of stem cell marker nucleostemin expression, these cells were identified as TNCs.

## Effect of irradiation and injection on tendon tissues

To specifically determine the fate of injected GFP tendon cells *in vivo*, it is essential to eliminate native tendon cells in wild-type mice. To achieve this, we irradiated adult C57BL/6J mice with 6 Gy using Gamma cell 40. The choice of 6 Gy dosage was based on previously published research analyzing the effect of a range of Gy dosages in whole body irradiation of mice monitored up to 30 days post-irradiation, with 6 Gy treated mice experiencing 100% survival rate with only mild or moderate radiation side effects [22]. We were able to replicate similar outcomes within our treated groups, with 100% survival and recovery. To evaluate our irradiation mode, a total of 3 normal and 3 irradiated wild type mice (six patellar tendons each) were analyzed. The irradiation treatment caused hair loss in wild-type mice and also changed the hair color from black to brown or even gray (**Fig 2A and 2D**). Live/dead cell viability assays were used to compare normal and irradiated tendon tissues, with results showing that irradiation eliminated the majority of live cells (red fluorescence, **Fig 2E and 2F**) compared to the normal mouse patellar tendons showing the majority of the tendon is populated with live cells (green fluorescence, **Fig 2B and 2C**). This experiment demonstrated the feasibility of the irradiation approach.

Next, we injected either isolated GFP-TSCs or GFP-TNCs into irradiated (6 Gy) mouse patellar tendons, as described in the methods and materials. One-week post-injection, the viability of injected GFP-tendon cells in mouse patellar tendon tissues was analyzed using live/

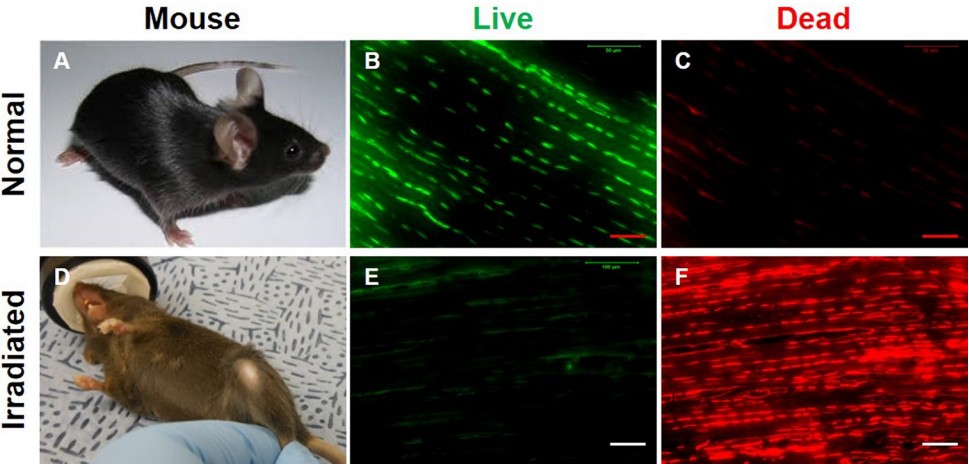

**Fig 2. Dead cells are predominant in irradiated mouse patellar tendons.** (**A**) A normal, untreated mouse has black hair. (**B, C**) Live/dead cell viability assay results show that normal mouse patellar tendon sections have more than 98% live cells as evidenced by green stained cell numbers and a few cells are dead cells which were stained by red fluorescence. (**D**) An irradiated (6 Gy) mouse exhibits loss of hair and change in hair color to brown or grey. (**E, F**) Live/dead cell viability assay results show that irradiation has killed most of the cells as evidenced by red fluorescence. n = 3 wild type untreated mice (**A-C**), and n = 3 irradiated wild type mice (**D-F**). Red bars: 50 μm; white bars: 100 μm.

dead cell viability assays, assessing 3 irradiated GFP-TSC mice and 3 irradiated GFP-TNC mice. Results showed clear viability of both GFP-TSCs and GFP-TNCs within tendon tissues (Fig 3A and 3B). Tendon tissues were also visualized for the level of GFP expression from injected GFP-tendon cells, with both TSCs (white arrows, Fig 3C) and TNCs (white arrows, Fig 3D) exhibiting similar levels of green fluorescence distributed throughout the tendon. Finally, GFP expression was verified independent of green fluorescence by staining with rabbit anti-GFP antibody, followed by the application of Cy3-conjugated goat anti-rabbit IgG antibody (Fig 3E–3H). The results demonstrated that the visible green fluorescence was due to GFP expression from injected GFP-TSCs (yellow arrows, Fig 3E) and GFP-TNCs (yellow arrows, Fig 3F). The merged images of GFP green fluorescence and Cy3 red fluorescence supports this conclusion (Fig 3G and 3H). Moreover, the results showed that the injections were done right beneath the paratenon (yellow arrow, Fig 3E). These injection experiments demonstrate the feasibility of injecting GFP-tendon cells (TSCs and TNCs) into mouse tendons, which remained viable and continued to express GFP after injection.

## The differential effects of MTR and ITR on GFP-TSC and GFP-TNC morphology *in vivo*

Mouse treadmill running experiments were initiated one week after the injection of either GFP-TSCs or GFP-TNCs into irradiated tendons of the mice, to allow treated tendons time to recover and to reconstitute with donor cells. Researchers have demonstrated that mice can have irradiation sickness in the first 7 days [22], thus this 1-week time period is essential for overall health and their ability to perform treadmill running. After ITR and MTR, tissues were collected and analyzed for changes in tissue architecture by H&E staining. The cell morphology of tendon cells exhibited a normal elongated shape typically found within tendon tissue within both normal (meaning healthy and untreated) tendon tissue (blue arrows, Fig 4B) and irradiation treated only tendon tissue (Fig 4D). However, the cell density in the paratenon of irradiation only tendon tissue was increased (red arrows, Fig 4D) compared to healthy/normal tissue.

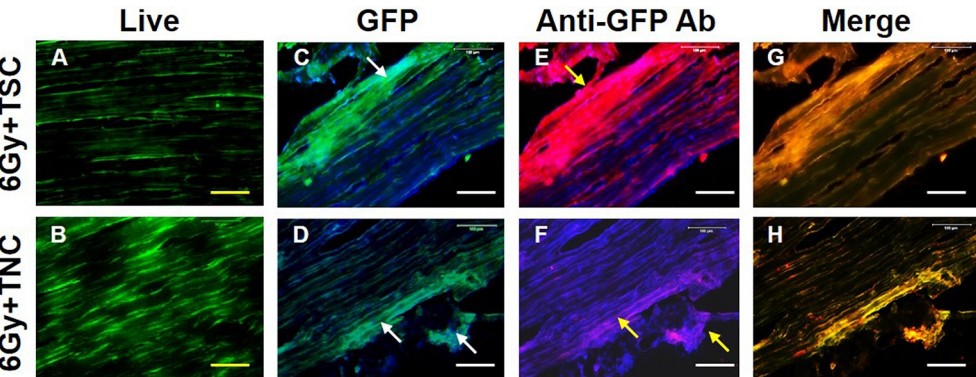

**Fig 3. Cells are viable after TSCs and TNCs are injected into irradiated patellar tendons. A, B**: Cell viability tested by live/dead assay kit. **C-D**: GFP expression from injected patellar tendons due to either GFP-TSCs (**C,** blue: DAPI staining) or GFP-TNCs (**D,** blue: DAPI staining). **E-F**: GFP-cell expression tested by immunostaining (blue: DAPI staining). **G-H**: Merged images of the **C, D** with **E, F** (only green with red for clarity). After one week, many cells are alive in the tendons injected with TSCs (green fluorescent cells in **A**) and TNCs (green fluorescent cells in **B**). GFP is highly expressed in tendon tissues injected either with TSCs (**C**) or TNCs (**D**) (white arrows in **C** and **D**). Immunostaining further demonstrated that these green fluorescent cells are GFP positive cells (yellow arrows in **E, F**) as evidenced by merged images expressing both green and red fluorescence (yellow fluorescence in **G, H**). For mice, n = 3 for both groups, meaning 3 irradiated GFP-TSC mice (top row) and 3 irradiated GFP-TNC mice (bottom row) were used in these experiments. Yellow bars: 50 μm; white bars: 100 μm.

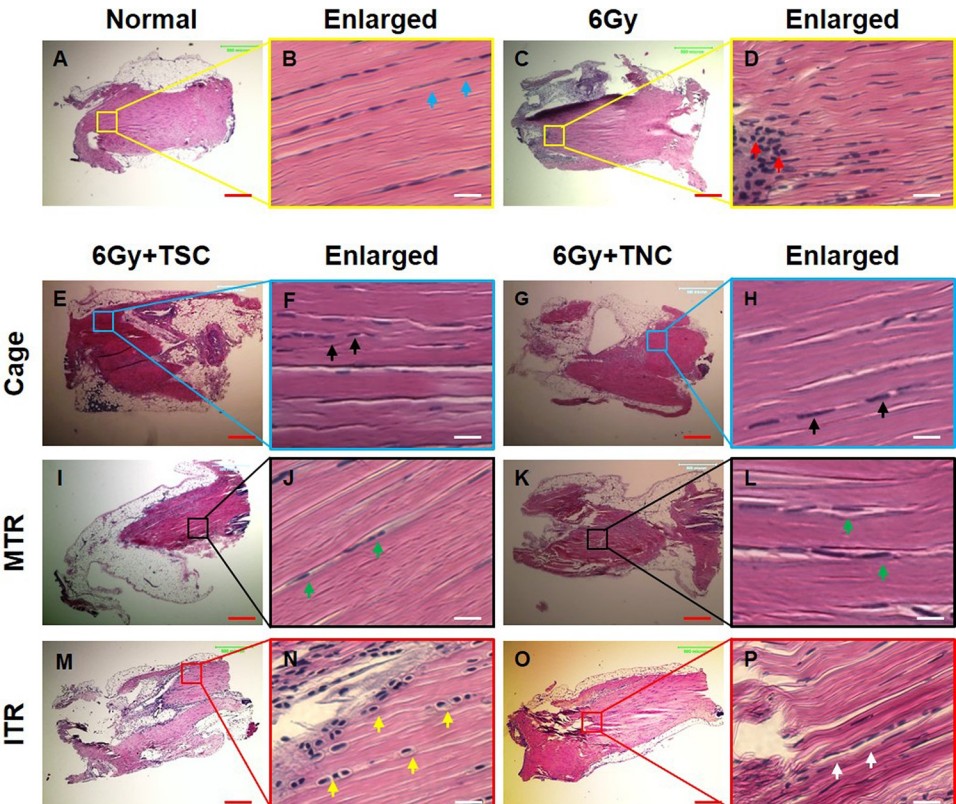

**Fig 4. Cell morphological changes in tendon tissue sections after irradiation, injection, and treadmill running.**
The H&E staining results show that normal cells (**A, B**) maintain the elongated shape (blue arrows in **B**) and irradiated cells (**C, D**) increase in density (red arrows in **D**). The cells either in TSCs injected tissues (**E, F**) or in TNCs injected tissues (**G, H**) do not change the shape in cage control condition (black arrows in **F, H**). Also, the cells either in TSCs injected tissues (**I, J**) or in TNCs injected tissues (**K, L**) do not change the shape after MTR (green arrows in **J, L**). However, the cells in tendon tissues injected with GFP-TSCs (**M, N**) assume a round shape (yellow arrows in **N**) after ITR. In contrast, the cells in TNCs-injected tissues (**O, P**) do not change shape (white arrows in **P**) after ITR. The images in the second and fourth columns are enlarged images of the boxes in the first and third columns, respectively. Red bars: 500 μm; white bars: 25 μm. MTR: 13 m/min, 50 min/day, 5 days a week for 3 weeks after a week's training; ITR: 13 m/min, 3 hrs, 4 hrs, and 5 hrs/day in the 2[nd], 3[rd], and 4[th] week respectively after a week's training. N = 6 untreated control mice were utilized in **A, B**, and n = 3 irradiation mice were utilized in **C, D**. For irradiated cage, MTR, and ITR mice, n = 3 mice (6 tendons) with GFP-TSCs, and n = 3 mice (6 tendons) with GFP-TNCs for each group were used.

## The differential responses in terms of morphology by MTR and ITR

Within our irradiation-injection mice, TSCs and TNCs exhibited different responses to MTR and ITR. Cell morphology within cage (mice irradiated and injected only) and MTR samples, regardless of whether tendons were injected with TSCs or TNCs, retained their normal elongated morphology (arrows, **Fig 4E–4L**). However, after ITR, tendons injected with GFP-TSCs exhibited a round shape (yellow arrows, **Fig 4N**), while tendons injected with GFP-TNCs and subjected to ITR did not (white arrows, **Fig 4P**). Thus, only TSCs, not TNCs, within tendon subjected to ITR responded by exhibiting a change in cell morphology indicative of chondrocytes.

## Chondrogenic differentiation within irradiated-injected animals subjected to MTR and ITR

To analyze this change in cell morphology further, we assessed the chondrogenic differentiation of TSCs and TNCs after MTR and ITR in irradiated-injected mice using Alcian blue

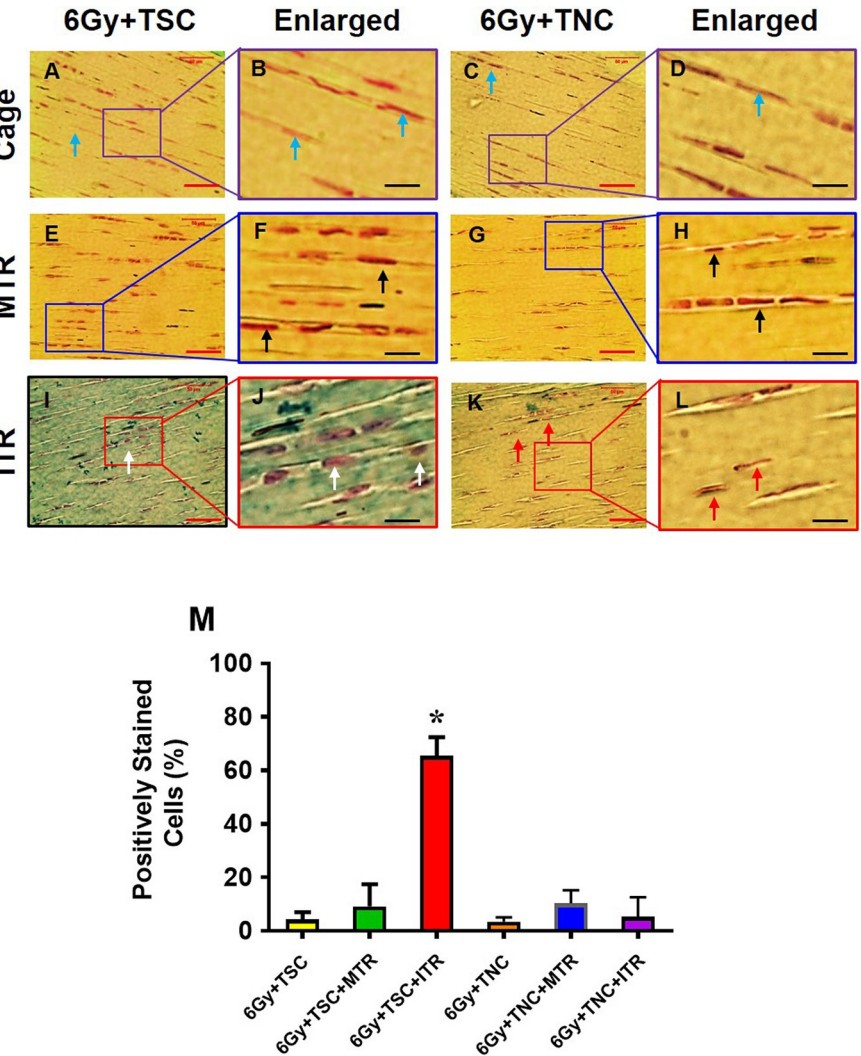

**Fig 5. ITR induces chondrogenic differentiation of TSCs in irradiated tendons.** Histochemical staining by Alcian blue and nuclear fast red shows that the cells in cage control tendons injected either with TSCs (**A, B**) or with TNCs (**C, D**) are still in elongated shape (blue arrows in **A, C**), and very few cells in cage control groups are positive for Alcian blue staining (**A-D**). Furthermore, both TSCs and TNCs in MTR tendons are still elongated shape (black arrows in **F, H**) and a few TSCs are positively stained by Alcian blue after MTR (**E, F**). However, some cells in ITR tendons are round shaped (white arrows in **I, J**), and many TSCs are positively stained by Alcian blue (**I, J**) after ITR. The TNCs are still in elongated shape (red arrows in **K, L**) and very little positive staining by Alcian blue (**K, L**). Semi-quantification of the chondrocyte staining shows that 66% of TSCs are positively stained with Alcian blue, and only 5% TNCs after ITR, and very few cells in cage control tendons and TSCs in MTR tendons are positively stained by Alcian blue (**M**). Graph bars represent the mean ± SD. *$p < 0.05$ compared to normal tendon. For each irradiated cage, MTR, and ITR group, n = 6 mice as described in Fig 5. Red bars: 50 μm; black bars: 12.5 μm.

staining. Cage control tendons without any treadmill running (*i.e.*, irradiation-injection only controls) exhibited a normal elongated shape and few cells stained with Alcian blue (blue arrows, **Fig 5A–5D**). Injected GFP-TSCs and GFP-TNCs in tendons within mice that underwent MTR also maintained a normal elongated shape, with the cellular matrix negatively stained for Alcian blue (black arrows, **Fig 5E–5H**). However, in ITR tendons with injected GFP-TSCs, cells exhibited a more rounded morphology indicative of chondrocytes (white arrows, **Fig 5I and 5J**), with an elevated level of Alcian blue staining (blue-green staining in **Fig 5I and 5J**). These results were in contrast to ITR tendons injected with GFP-TNCs

exhibiting very little change in either cell morphology by maintaining an elongated cell shape and by producing minimal Alcian blue staining (red arrows, **Fig 5K and 5L**). Semi-quantification of chondrocyte staining showed that at least 66% of injected TSCs after ITR were positively stained with Alcian blue, but only 5% of TNCs exhibited staining after ITR (**Fig 5M**). Furthermore, TSCs and TNCs in MTR samples also exhibited low levels of Alcian blue staining similar to cage control samples (**Fig 5M**).

### Differential effects of MTR and ITR on non-tenocyte related gene expression in GFP-TSCs and GFP-TNCs injected tendons *in vivo*

We analyzed the gene expression in GFP-TSC and GFP-TNC injected patellar tendon tissue followed by MTR and ITR regimens by focusing on LPL (an adipocyte related gene), Runx-2 (a bony tissue-specific gene), and SOX-9 (a cartilage-specific gene) using qRT-PCR analysis. Control tissues refer to either GFP-TSC injected cage controls (**Fig 6A and 6B**) or GFP-TNC injected cage controls (**Fig 6C and 6D**) without treadmill running. TSCs responded to MTR by increasing the expression of tenocyte-related genes collagen I and tenomodulin, with little change in the expression of non-tenocyte genes; however, TSCs responded to ITR by elevating the expression of non-tenocyte related genes (**Fig 6A and 6B**). Both MTR and ITR resulted in TNCs exhibiting increased tenocyte-related gene expression (**Fig 6C**); however, very little effect was seen within non-tenocyte markers (**Fig 6D**). Thus, only GFP-TSC injected tendons showed signs of non-tenocyte related differentiation in response to ITR.

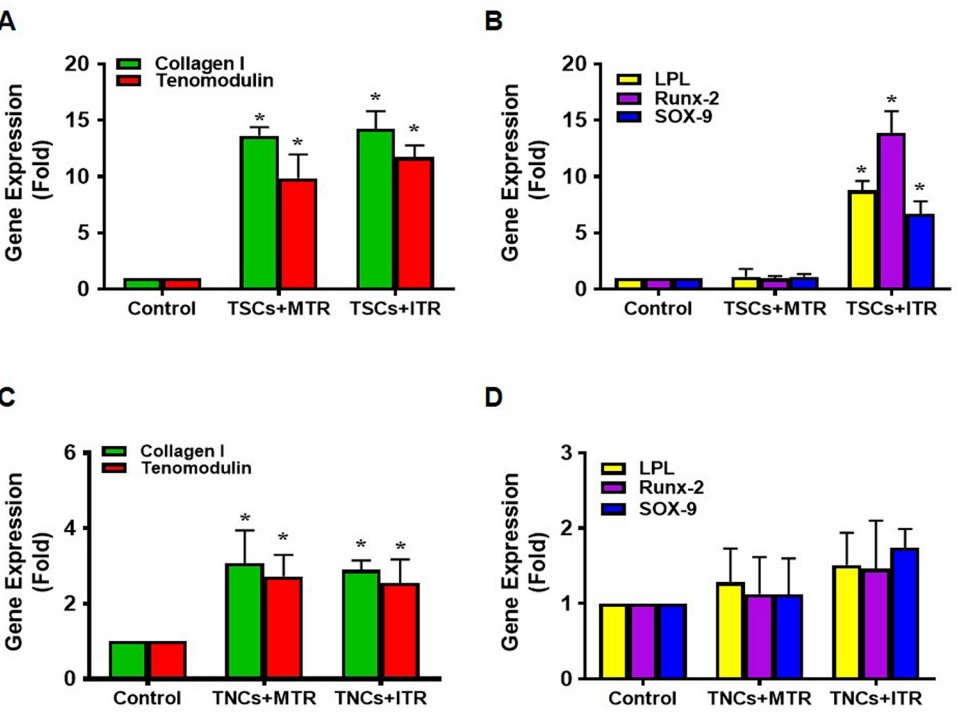

**Fig 6. ITR induces significant non-tenocyte related gene expression in TSCs of irradiated tendons.** Both MTR and ITR induce significant expression of tenocyte related genes, collagen I and tenomodulin in TSCs (**A**), however, only ITR induces non-tenocyte related gene expression of LPL, Runx-2, and SOX-9 in TSCs (**B**). MTR and ITR also significantly induce collagen I and tenomodulin (**C**) without the induction of non-tenocyte related gene expression in TNCs (**D**). For each group sample, n = 6 tendons were used for analysis. Graph bars represent the mean ± SD. *p < 0.05 compared to each cage control group.

Immunostaining of tendon tissue sections revealed that the ITR regimen induced differentiation of GFP-TSCs into non-tenocytes as evidenced by positive staining of SOX-9 (chondrocyte marker), Runx-2 (osteocyte marker), and PPARγ (adipocyte marker) (**Fig 7G–7I**), in comparison to cage controls with irradiation-injection only (**Fig 7A–7C**) and GFP-TSC injected mice with MTR (**Fig 7D–7F**). In GFP-TNC injected tissues, both MTR and ITR

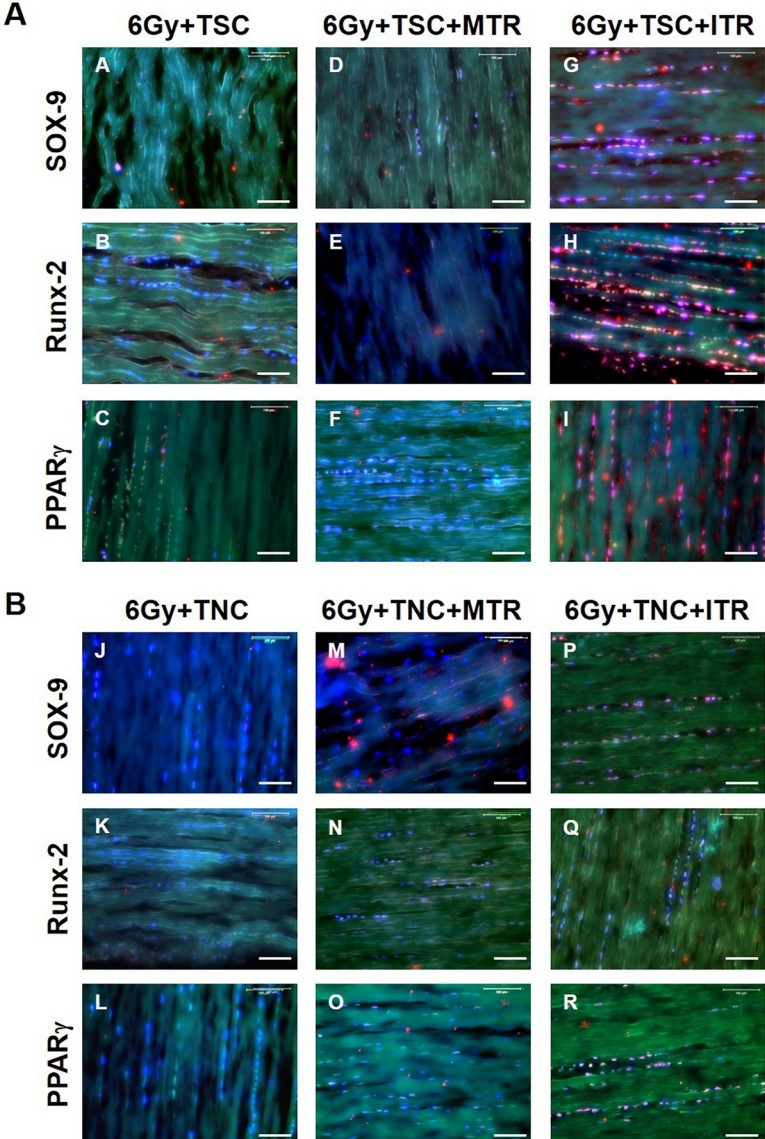

**Fig 7. ITR induces non-tenocyte differentiation of TSCs in irradiated tendons determined by immunostaining.**
Top panel (**A**): **A-C**: TSCs-transplanted, irradiated tendons without running; **D-F**: TSCs-transplanted, irradiated tendon after MTR; **G-I**: TSCs-transplanted, irradiated tendon after ITR; Bottom panel (**B**): **J-L**: TNCs-transplanted, irradiated tendon without running; **M-O**: TNCs-transplanted, irradiated tendon after MTR; **P-R**: TNCs-transplanted, irradiated tendons after ITR. There is no positive staining in TSCs-transplanted, irradiated tendons (**A-C**) and TNCs-transplanted (**J-L**) without running for all three non-tenocyte markers including SOX-9 (**A, J**), Runx-2 (**B, K**), and PPARγ (**C, L**). A few positively stained cells are found in TNCs-transplanted, irradiated tendons after ITR (**P-R**), and in TSCs-transplanted, irradiated tendons after MTR (**D-F**). However, a high percentage TSCs (**G-I**) are positively stained after ITR indicating the abundant presence of chondrocytes (**G**), osteocytes (**H**) and adipocytes (**I**) in mouse tendon tissues injected with TSCs that have undergone ITR. Three colors–green, red, and blue (DAPI) are merged in each of the images. Also, for each group sample, n = 6 tendons were used for analysis. Bars: 100 μm.

samples exhibited few cells stained for all three markers (**Fig 7M–7R**) compared to control samples (**Fig 7J–7L**). The low amount of positively stained cells seen in GFP-TNC injected tendons may be attributed to the presence of a small number of GFP-TSCs that remained from the primary culture of tendon cells initially isolated from GFP-expressing mice. Collectively, these results demonstrate that only ITR, but not MTR, is able to induce non-tenocyte differentiation in the tendon injected with GFP-TSCs.

## Discussion

This study showed that GFP-TSC injected patellar tendons within ITR mice exhibited rounding of cells, elevated chondrogenic differentiation, and elevated gene expression and immunostaining of non-tenocyte markers, indicative of some of the characteristics of tendinopathy such as the rounding of cell nuclei and acquisition of chondrocyte phenotypes [4, 6, 7, 9, 28]. These changes due to ITR were lacking in GFP-TNC injected tendons, and in MTR mice overall. Taken together, these data suggest that resident TSCs, but not TNCs, contribute to the onset of tendinopathy development through aberrant differentiation.

To specifically study the role of TSCs and TNCs in tendinopathy within an *in vivo* environment, we utilized an irradiation-injection mouse model, unlike previous *in vivo* models that evaluated these cell types in an environment comprised of possibly many other factors. To the best of our knowledge, this is the first *in vivo* study to validate the contribution of TSCs in the onset of tendinopathy development by their aberrant differentiation.

Previous studies including ours have utilized isolated TSCs after mechanical loading to study differentiation of TSCs *in vitro* [11, 13, 29]. However, a reliable *in vivo* method is needed for tracking the fate of TSCs. Using our irradiation and injection model, we were able to directly visualize the fate of cells under mechanical loading conditions *in vivo*, thereby providing a superior and reliable method for tracking the fate of each tendon cell type. This new approach is effective, with our results showing that ITR induces aberrant non-tenocyte differentiation of TSCs *in vivo*, while TNCs remain unaffected under the same mechanical overloading condition.

Abnormal tendinopathic tissues are characterized by hypercellularity, hypervascularity, and acquisition of chondrocyte phenotype, which are markers of late stage tendinopathy and also are signs of an active cell-mediated process in degenerative tendons [5, 7]. The presence of stem cells in tendon with self-renewal and multi-differentiation potential is well-established in human and animal tendons [20, 30, 31]. Under the influence of abnormal environmental cues such as mechanical overloading, these cells are capable of differentiating into chondrocytes, adipocytes, and osteocytes which can initiate the formation of non-tendinous tissues within tendon. The ability of isolated TSCs to differentiate into either TNCs or into non-tendinous cells dependent upon the level of loading *in vitro* has been well established [11, 13, 16], with the caveat that any outside influences that may have an impact on results are excluded within an *in vitro* model.

As with *in vivo* models, two separate mechanisms are possible to explain the presence of non-tendinous tissues within overloaded tendons. Tendinopathic phenotypes are either due to the migration of cells to the tendon tissue, or due to resident cells within tendon tissue responding to overloading by differentiating into non-tenocytes. The migration of bone marrow mesenchymal stem cells (BMSCs) can be affected by mechanical strain [10, 32]. In this study, we took a step further to investigate the fate of injected GFP-TSCs or GFP-TNCs, devoid of other cell populations, in tendon under different mechanical loading conditions (MTR and ITR) *in vivo*. We believe that this approach is more reliable and superior for tracking the differentiation of isolated/resident TSCs after mechanical loading.

In contrast to ITR, MTR induced tenocyte differentiation of TSCs in TSC-injected mice led to an anabolic environment characterized by the increased gene expressions of collagen I and tenomodulin. This finding is in agreement with our previous *in vitro* data that showed mechanical stretching of TSCs at a moderate level significantly increased only the expression of tenocyte-related gene collagen I, but not the expression of non-tenocyte related genes PPARγ, collagen II, SOX-9, and Runx-2, which are markers of adipocytes, chondrocytes, and osteocytes respectively [11]. This finding is also in agreement with our previous *in vitro* and *in vivo* study with isolated TSCs from patellar and Achilles tendons of mice subjected to MTR, which showed that two tenocyte-related genes, collagen I and tenomodulin were upregulated by MTR [13]. Hence, MTR may promote tendon repair and remodeling by promoting differentiation of TSCs into TNCs.

Whole body irradiation was selected as a viable option due to previously published research showing the efficacy and safety of this technique [22, 23]. While irradiation may seem like a harsh treatment, great care was taken to analyze previously published data to select a viable dosage for our experiments, as well as continual monitoring of animal health and welfare within this study. While our results prove the feasibility of cell tracking using this approach, caution should be exercised in lessening the impact of irradiation on animals. In future studies, alternative methods of irradiation such as conditional cell ablation, a Cre-mediated lineage deletion technique in mice [33, 34], that has significantly less impact on the welfare of small animals may be adopted. However, the development of such a mouse model for the conditional deletion of specific tendon cell lineages would require further research, time, and resources.

This study has a few limitations. Pure cell populations could be isolated from tendon cell cultures using fluorescence-activated cell sorting with antibodies to CD18, CD45, and CD90 [35]. Furthermore, TSCs and TNCs could be further distinguished and verified by using other stem cell markers such as Oct-4, Nanog, and SSEA-4 [20], and additional markers for tenocytes. Also, further evaluation of injected GFP-TNCs and newly generated TNCs due to GFP-TSC differentiation must be included within both ITR and MTR groups. We used collagen I and tenomodulin as markers of TNCs for an initial investigation. Additional research is needed to fully evaluate the role of TNCs within MTR, and the effect on tendon healing, repair, and homeostasis. Finally, while this study found that short term ITR (3 weeks) induces the onset of tendinopathy development by causing differentiation of TSCs into non-tenocytes in mice, a future study is warranted to define the full development of tendinopathy at both the cellular, tissue, and functional levels after long term ITR.

In conclusion, this study used an irradiation-injection mouse model to investigate the differential mechanobiological responses of TSCs to short term MTR and ITR. It showed that while MTR promotes TSCs to differentiate into TNCs and may be used as a therapy to promote tendon homeostasis, repair, and remodeling of tendon tissue after injury, ITR induces aberrant differentiation of TSCs into non-tenocytes, thus likely contributing to the onset of tendinopathy development.

## Supporting information

**S1 Fig. Schematic of experimental design and animal numbers of the study.**
(TIF)

## Acknowledgments

We thank Dr. Bhavani P Thampatty for assistance in the preparation of this manuscript.

## Author Contributions

**Conceptualization:** Jianying Zhang, MaCalus V. Hogan, James H-C. Wang.

**Data curation:** Jianying Zhang, Daibang Nie, Arthur McDowell.

**Formal analysis:** Daibang Nie, Kelly Williamson.

**Funding acquisition:** James H-C. Wang.

**Methodology:** Jianying Zhang, Daibang Nie, Arthur McDowell.

**Project administration:** James H-C. Wang.

**Supervision:** MaCalus V. Hogan, James H-C. Wang.

**Validation:** Jianying Zhang, Kelly Williamson.

**Writing – original draft:** Jianying Zhang.

**Writing – review & editing:** Jianying Zhang, Kelly Williamson, James H-C. Wang.

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
