## [Decision Letter · Decision Letter 0]

20 Nov 2020

PONE-D-20-34875

Moderate and intensive mechanical loading differentially modulate the phenotype of tendon stem/progenitor cells in vivo

PLOS ONE

Dear Dr. Wang,

Thank you for submitting your manuscript to PLOS ONE. After careful consideration, we feel that it has merit but does not fully meet PLOS ONE’s publication criteria as it currently stands. Therefore, we invite you to submit a revised version of the manuscript that addresses the points raised during the review process.

Although some interest this manuscript needs to be consistently amended following all the criticisms raised by the two referees and reported below.

In particular results are over interpreted, mainly because the Authors did not evaluate other parameters including flogosis, immunomodulation and others. 

Several other issues must be addressed including the English style and grammar.

We look forward to receiving your revised manuscript.

Kind regards,

Gianpaolo Papaccio, M.D., Ph.D.

Academic Editor

PLOS ONE

Journal Requirements:

"This work was supported in part by the National Institute of  Arthritis and Musculoskeletal and Skin Diseases (NIAMS), NIH (https://www.niams.nih.gov/) under awards AR065949 and AR070340 (JHW). The funders had no role in study design, data collection and analysis, decision to publish, or preparation of the manuscript."

3. Please include your tables as part of your main manuscript and remove the individual files. Please note that supplementary tables (should remain/ be uploaded) as separate "supporting information" files

Reviewers' comments:

Reviewer's Responses to Questions

**Comments to the Author**

1. Is the manuscript technically sound, and do the data support the conclusions?

Reviewer #1: Partly

Reviewer #2: Partly

2. Has the statistical analysis been performed appropriately and rigorously? 

Reviewer #1: Yes

Reviewer #2: Yes

3. Have the authors made all data underlying the findings in their manuscript fully available?

Reviewer #1: Yes

Reviewer #2: Yes

4. Is the manuscript presented in an intelligible fashion and written in standard English?

Reviewer #1: Yes

Reviewer #2: No

5. Review Comments to the Author

Reviewer #1: In this papers Authors evaluated the effect of TSCs and TNCs injection in irradiated mice after MTR and ITR.

The study is interesting but some concerns need to be addresses.

Authors should submit a higher resolution figure 3. Additionally, nuclear staining with DAPI or HOECHST is required to clarify the presence of cells.

The same observation applies to figure 7.

In the discussion Authors state that in their study they demonstrate TSCs are responsible for the development of tendinopathies during ITR through aberrant differentiation. However, Authors exclusively compare the differentiation capacities of TSCs and TNCs during MTR and ITR. So the conclusions need to be revised. To confirm what they stated, it would also be necessary to evaluate hypercellularity, hypervascularity and expression of inflammatory agents.

Reviewer #2: The manuscript is very interesting; the figures are good. The Authors demonstrated that GFP-TSC injected patellar tendons within ITR mice exhibited rounding of cells, elevated chondrogenic differentiation, and elevated gene expression and immunostaining of non-tenocyte markers. Although this, the authors should address some concerns. The results are overinterpreted when the authors state that resident TSCs, but not TNCs, contribute to the development of ITR-induced tendinopathy through aberrant differentiation. In my opinion in order to better characterize the type of tendinopaty, the authors must evaluate other parameters such as inflammation, immunomodulating factors, proliferation and tendon disintegration. In fact, in the phase of reactive or acute tendinopathy there is an accumulation in loco of cytokines and immunomodulating factors that cause an initial inflammation with cell proliferation. The transition from the reactive tendinopathy phase to that of degenerative tendonitis corresponds to the failed healing phase (failure of the self-healing process) which causes tendon disintegration. The tendon, at the end of this process, will have degenerated portions that become unable to develop tensile strength. Therefore, the Authors must change the conclusions or add experiments that confirm their hypotheses.

In addition, the manuscript must be revised by a native English speaker.

6. PLOS authors have the option to publish the peer review history of their article (what does this mean?). If published, this will include your full peer review and any attached files.

Reviewer #1: No

Reviewer #2: No

---

## [Author Response · Author response to Decision Letter 0]

1 Dec 2020

Editor comments and response

In particular results are over interpreted, mainly because the Authors did not evaluate other parameters including flogosis, immunomodulation and others. 

Response: We have clarified this issue in

the revision.

Several other issues must be addressed including the English style and grammar.

Response: We have addressed all the issues raided by the reviewers.

Amendment of funding statement.

Response: We have amended the statement and included in the cover letter.

Insert table in the text.

Response: We have included Table in the text.

Reviewer #1

General Comment: In this papers Authors evaluated the effect of TSCs and TNCs injection in irradiated mice after MTR and ITR. The study is interesting but some concerns need to be addresses

Response: Thank you for your comments. 

Specific Comments

1. Authors should submit a higher resolution figure 3. Additionally, nuclear staining with DAPI or HOECHST is required to clarify the presence of cells.

The same observation applies to figure 7.

Response: We have provided new figures 3 and 7, with a higher resolution. DAPI staining has been added to show the presence of cells, as requested. 

2. In the discussion Authors state that in their study they demonstrate TSCs are responsible for the development of tendinopathies during ITR through aberrant differentiation. However, Authors exclusively compare the differentiation capacities of TSCs and TNCs during MTR and ITR. So, the conclusions need to be revised. To confirm what they stated, it would also be necessary to evaluate hypercellularity, hypervascularity and expression of inflammatory agents.

Response: The purpose of this study was to determine the differential mechanobiological responses between TSCs and TNCs. We found that in response to ITR but not MTR, TSCs but not TNCs were able to differentiate into non-tenocytes. Based on this finding, we speculated that TSCs may be responsible for the development of tendinopathy. But keep in mind that since the mechanical overloading (ITR) was relatively short (3 weeks), it is likely that the state of tendinopathy was just in an early developing stage. To achieve a full-blown development of tendinopathy, which exhibits the said characteristics – hypercellularity, hypervascularity, and expression/production of inflammatory agents, a much longer time period is required. Indeed, maturation of tendinopathy takes years of mechanical overloading placed on the tendon. We have revised the related text to make this point clear. 

Reviewer: 2 

General Comment: The manuscript is very interesting; figures are good. The Authors demonstrated that GFP-TSC injected patellar tendons within ITR mice exhibited rounding of cells, elevated chondrogenic differentiation, and elevated gene expression and immunostaining of non-tenocyte markers. Although this, the authors should address some concerns.

 Response: Thank you for your summary of this study. 

Specific Comments

1. The results are overinterpreted when the authors state that resident TSCs, but not TNCs, contribute to the development of ITR-induced tendinopathy through aberrant differentiation. In my opinion in order to better characterize the type of tendinopathy, the authors must evaluate other parameters such as inflammation, immunomodulating factors, proliferation and tendon disintegration. In fact, in the phase of reactive or acute tendinopathy there is an accumulation in loco of cytokines and immunomodulating factors that cause an initial inflammation with cell proliferation. The transition from the reactive tendinopathy phase to that of degenerative tendonitis corresponds to the failed healing phase (failure of the self-healing process) which causes tendon disintegration. The tendon, at the end of this process, will have degenerated portions that become unable to develop tensile strength. Therefore, the Authors must change the conclusions or add experiments that confirm their hypotheses.

In addition, the manuscript must be revised by a native English speaker.

Response: As explained in our response to reviewer #1 above, our study focused on how TSCs respond differently from TNCs. We showed that TSCs but not TNCs were able to differentiate into non-tenocytes under a short term, or 3 weeks of mechanical overloading condition. We suggest that our results are from early developing stage of tendinopathy, whereas what the reviewer described are characteristics of later stage of tendinopathy, mostly in humans. We have clarified the related text in the revision. 

In terms of the need for a native English speaker for revision of this manuscript, the authors are not clear about the specific aspects of English usage problems which are referred in the reviewer’s comment. Indeed, there are a few native English speakers (KW, AM, MH) who participated in drafting this manuscript. Nevertheless, we have closely revised the entire manuscript.

---

## [Decision Letter · Decision Letter 1]

14 Dec 2020

Moderate and intensive mechanical loading differentially modulate the phenotype of tendon stem/progenitor cells in vivo

PONE-D-20-34875R1

Dear Dr. Wang,

We’re pleased to inform you that your manuscript has been judged scientifically suitable for publication and will be formally accepted for publication once it meets all outstanding technical requirements.

Kind regards,

Gianpaolo Papaccio, M.D., Ph.D.

Academic Editor

PLOS ONE

Additional Editor Comments (optional):

Reviewers' comments:

Reviewer's Responses to Questions

**Comments to the Author**

1. If the authors have adequately addressed your comments raised in a previous round of review and you feel that this manuscript is now acceptable for publication, you may indicate that here to bypass the “Comments to the Author” section, enter your conflict of interest statement in the “Confidential to Editor” section, and submit your "Accept" recommendation.

Reviewer #1: All comments have been addressed

Reviewer #2: All comments have been addressed

2. Is the manuscript technically sound, and do the data support the conclusions?

Reviewer #1: (No Response)

Reviewer #2: Yes

3. Has the statistical analysis been performed appropriately and rigorously? 

Reviewer #1: (No Response)

Reviewer #2: Yes

4. Have the authors made all data underlying the findings in their manuscript fully available?

Reviewer #1: (No Response)

Reviewer #2: Yes

5. Is the manuscript presented in an intelligible fashion and written in standard English?

Reviewer #1: (No Response)

Reviewer #2: Yes

6. Review Comments to the Author

Reviewer #1: (No Response)

Reviewer #2: (No Response)

7. PLOS authors have the option to publish the peer review history of their article (what does this mean?). If published, this will include your full peer review and any attached files.

Reviewer #1: No

Reviewer #2: No

---

## [Editor Report · Acceptance letter]

16 Dec 2020

PONE-D-20-34875R1 

Moderate and intensive mechanical loading differentially modulate the phenotype of tendon stem/progenitor cells *in vivo*

Dear Dr. Wang:

I'm pleased to inform you that your manuscript has been deemed suitable for publication in PLOS ONE. Congratulations! Your manuscript is now with our production department. 

Kind regards, 

on behalf of

Prof. Gianpaolo Papaccio 

Academic Editor

PLOS ONE